# Using digital tools in clinical, health and social care research: a mixed-methods study of UK stakeholders

Sophie Clohessy [1], Theodoros N Arvanitis [2], Umer Rashid,[1] Carly Craddock,[3,4] Mark Evans,[3,4] Carla T Toro [5], Mark T Elliott [1,6]

¹WMG, University of Warwick, Coventry, UK
²School of Engineering, University of Birmingham, Birmingham, UK
³National Institute for Health Research Clinical Research Network West Midlands, Birmingham, UK
⁴The Royal Wolverhampton NHS Trust, Wolverhampton, UK
⁵Warwick Medical School, University of Warwick, Coventry, UK
⁶School of Sport, Exercise and Rehabilitation Sciences, University of Birmingham, Birmingham, UK

**Correspondence to**
Dr Mark T Elliott;
m.elliott.3@bham.ac.uk

## ABSTRACT

**Objective** The COVID-19 pandemic accelerated changes to clinical research methodology, with clinical studies being carried out via online/remote means. This mixed-methods study aimed to identify which digital tools are currently used across all stages of clinical research by stakeholders in clinical, health and social care research and investigate their experience using digital tools.

**Design** Two online surveys followed by semistructured interviews were conducted. Interviews were audiorecorded, transcribed and analysed thematically.

**Setting, participants** To explore the digital tools used since the pandemic, survey participants (researchers and related staff (n=41), research and development staff (n=25)), needed to have worked on clinical, health or social care research studies over the past 2 years (2020–2022) in an employing organisation based in the West Midlands region of England (due to funding from a regional clinical research network (CRN)). Survey participants had the opportunity to participate in an online qualitative interview to explore their experiences of digital tools in greater depth (n=8).

**Results** Six themes were identified in the qualitative interviews: 'definition of a digital tool in clinical research'; 'impact of the COVID-19 pandemic'; 'perceived benefits/drawbacks of digital tools'; 'selection of a digital tool'; 'barriers and overcoming barriers' and 'future digital tool use'. The context of each theme is discussed, based on the interview results.

**Conclusions** Findings demonstrate how digital tools are becoming embedded in clinical research, as well as the breadth of tools used across different research stages. The majority of participants viewed the tools positively, noting their ability to enhance research efficiency. Several considerations were highlighted; concerns about digital exclusion; need for collaboration with digital expertise/clinical staff, research on tool effectiveness and recommendations to aid future tool selection. There is a need for the development of resources to help optimise the selection and use of appropriate digital tools for clinical research staff and participants.

## INTRODUCTION

Digital tools are increasingly used across stages of clinical research from recruitment (eg, social media platforms) to data collection (eg, online survey platforms, online

## STRENGTHS AND LIMITATIONS OF THIS STUDY

⇒ Mixed-method study to explore experiences of using digital tools in clinical research, using participants from a variety of roles; both research and research and development staff.
⇒ Survey and qualitative questions codesigned with an experienced stakeholder group.
⇒ It is likely that participants were motivated to participate due to an underlying interest in digital tools, consequently some perspectives may not have been included in this study.
⇒ The study was limited to exploring the views of stakeholders employed by organisations based in the West Midlands, UK.

workshops)[1–4], with the applicability of digital tools within clinical research ever widening . Digital tools can be defined as an alternative to paper-based methods, that is, IT-based or an online platform that aids any aspect of the research study. The COVID-19 pandemic accelerated changes to clinical research methodology, social distancing forced study teams to transition from conducting research activities in person to predominantly online and/or via remote means.[1 5 6] Documented challenges associated with rapid adoption of digital tools in clinical research include; staff/participant lack of familiarity with or limited access to digital tools and rapid changes to protocols/ethics applications to ensure compliance.[5]

A recent report by the UK government (see 'The Future of Clinical Research Delivery: 2022 to 2025 implementation plan') sets out a future vision of research delivery which includes data enabled trials (ie, using electronic health records) and digital tools.[7] Given that digital tools are now commonplace and set to be firmly embedded within clinical research, there is a need to explore people's experiences using digital tools; what has worked well and not so well in order to understand their value, relevance and use in

**Table 1** Eligibility criteria by survey type

| Survey type | Participant group | Eligibility criteria |
|---|---|---|
| Researchers and related staff | People working on clinical research projects either in a research capacity or operational support. | Aged 18 years or older; employing organisation based in West Midlands region of the UK; working on clinical research projects over the past 2 years (2020–2022) |
| Research and development (R&D) staff | People working within a hospital's R&D (or similar) department, with a role to support or coordinate clinical research taking place within, or in partnership with the hospital. | Aged 18 years or older; hospital must be a research active NHS Trust in the West Midlands region of the UK; working in a hospital's R&D (or similar) department over the past 2 years (2020–2022) |

NHS, National Health Service.

future clinical research. A recent mixed-method study by Blatch-Jones *et al*[8] explored which digital tools were used by stakeholders within recruitment and retention in clinical research in the UK as well as investigating experiences using these tools. While previous research has investigated UK stakeholder's opinions on digital tool adoption for recruitment and retention, this has left other research stages understudied. It is evident there is a need to explore experiences of digital tools across the entire breadth of research stages (eg, informed consent, set-up, data collection, intervention delivery, management of research studies) as well as investigating the impact of the COVID-19 pandemic on the switch to digital tools for facilitating clinical research. Indeed, a particular focus of this study was on the use and experience of digital tools in the context of the rapid switch to digital methods during the COVID-19 pandemic in the UK between 2020 and 2022.

The aim of this study was to explore examples of digital tools used by stakeholders across the West Midlands geographical region in clinical, health and social care research.

## METHODS

This study comprised two main components: Phase 1: Online surveys to understand the overall picture of digital tool adoption across the region, with key stakeholder groups (see table 1 for the definition of these groups): (1) researchers and related staff, (2) research and development (R&D) staff and (3) participants and carers (due to limited responses (n=4) from the participant and carers group, no further data analysis took place). Phase 2: Online survey participants were given the opportunity to participate in qualitative interviews to build on survey answers, exploring their experiences with digital tools in further depth. The survey and qualitative questions were codesigned with an experienced stakeholder group consisting of three individuals working in senior positions within clinical research based in the West Midlands. The stakeholder group met once during the project on Microsoft Teams (February 2022), the research team presented an overview of the project and survey questions and the steering group offered verbal feedback. The stakeholder group members were emailed a draft of qualitative

interview materials in August 2022 and provided feedback via email on the qualitative questions.

### Patient and public involvement
No patients and/or public were involved in the design or conduct or reporting or dissemination plans of this research.

### Phase 1: online surveys
*Participants*
Two surveys were aimed at the following participant groups, with eligibility criteria shown in table 1.

*Recruitment*
i. Researchers and related staff survey: Participants were recruited primarily using databases of clinical, health and social care research studies held by the National Institute of Health Research CRN West Midlands (NIHR, a virtual organisation, is the research arm of the National Health Service. The NIHR CRN is made up of 15 local networks across England which helps patients, the public and health and care organisations to participate in high-quality research). An initial list of relevant projects was obtained by NIHR staff using the Open Data Platform that transforms the data held in the NIHR's Central Portfolio Management System (CPMS) into a usable form, allowing it to be tabled, filtered and organised by various metrics. Projects were selected based on the following criteria: clinical, health or social care projects supported by the NIHR CRN West Midlands, which had been completed or were ongoing within the previous 3 years - although we were primarily interested in including studies which had used digital tools between the years 2020–2022 (since beginning of COVID-19), studies were included from the past 3 years (2019–2022) to ensure a suitable number of studies were included in the survey. Any recruitment activity during the four financial years 2019/2020–2022/2023 was filtered to include only those studies opened since 1 June 2019. Additional data were sourced from CPMS and the Local Portfolio Management System, which functions as a secure data storage platform for the records of study approvals, delivery and site activity at a regional level. An email invitation and survey link were then

distributed to the lead researcher or main contact for the selected research studies (approximately 400 contacts in total).

ii. R&D survey: A total of 26 NHS Trust R&D departments in the West Midlands were contacted by NIHR staff via email invitation and survey link.

For both surveys, to further widen the opportunity to participate, a short study description and survey links were advertised on Twitter via the NIHR CRN West Midlands account. After a period of 2 weeks, a follow-up reminder email was sent to all contacts via the same mechanism as the initial survey invitation. Both surveys were live for one month in total. We anticipated approximately 50 people would complete each survey, however, after the first data collection period (27 June 2022–26 July 2022), participant numbers were lower than expected (35 for research and related staff survey and 15 for R&D); therefore, a second phase of recruitment took place (14 October 2022–15 November 2022). An email prompt was sent to all contacts via the same mechanism as the initial survey invitation.

### Procedure

Participants accessed surveys via a hyperlink. The surveys were created and hosted on Qualtrics, a secure online survey platform (https://www.qualtrics.com/uk/), which is used for creating surveys with in-depth response formats and allows for sophisticated methods of distribution and data management.[9] Before taking part, all participants read an information sheet and ticked a consent form (embedded into the Qualtrics platform) to agree to their participation. After survey completion, participants were presented with a debriefing sheet which repeated the objectives of the study and information about how the survey data would be stored. Before answering any questions, participants were offered a definition of a digital tool and examples of digital tools (see online supplemental table S1). The definition varied between the two surveys to account for tools used in a variety of contexts. All survey participants were invited to participate in a qualitative interview study to explore their thoughts towards digital tools in further detail.

### Researchers and related staff survey: procedure overview

Participants were redirected to the end of the survey if they did not meet the eligibility criteria (see table 1) determined via initial screening questions. A further screening question determined if participants had used digital tools in their research studies. Participants who answered 'yes' to using digital tools in their clinical, health or social care research were then directed to questions which focused on participants' experience of what they perceived to be most effective and least effective digital tools used in their role (blocks presented in a randomised order, questions were the same in each block) (see online supplemental table S2 for question themes and online supplemental materials 2a for online survey questions).

### R&D staff survey: procedure overview

Eligible participants answered three blocks of questions about their perceived most effective digital tool used in (1) set-up, (2) recruitment and (3) management of clinical research studies (blocks were presented in a randomised order, questions were the same in each block). If participants did not have a digital tool example for set-up, recruitment or management they skipped this block of questions and were redirected to the next part of the survey (see online supplemental table S2 for question themes and online supplemental materials 2b for online survey questions).

### Data analysis

A mixed approach to analysis was undertaken for both surveys. Multiple choice and scale answers were analysed using frequencies and percentages; free text answers were analysed thematically. Results from both surveys were analysed, discussed and verified with the project team.

### Phase 2: qualitative interviews
#### Recruitment

During the online survey, participants were invited to take part in the qualitative interviews and expressed their interest by providing their name, job role and email address.

#### Procedure

All interested participants were sent an electronic information and consent sheet via email and a mutually convenient time and date was arranged via Doodle poll (https://doodle.com/free-poll). Participants returned consent via email; this was also confirmed verbally at the beginning of each interview. Semistructured interviews were undertaken by the research fellow (author, SC) via Microsoft Teams, lasting between 45 and 60 min (conducted between 20 October 2022–9 November 2022). The interview topic guides (see online supplemental table S3 for an overview and online supplemental materials 3a and 3b for full interview questions) were informed by the online survey; participants were informed that their survey answers would be discussed in greater detail during the qualitative interviews (referenced on the consent form). All interviews were recorded and transcribed using the built-in functionality in Microsoft Teams. Data collection was stopped at the point of data saturation; this was defined as the point of which no new themes were observed.

#### Data analysis

Transcripts were anonymised and cross-checked for accuracy by the first author (SC) and research assistant (UR) against the recordings. Interview transcripts were then uploaded into NVivo qualitative analysis software (V.12, QSR International, Massachusetts, USA). A thematic analysis[10] was applied, using an inductive approach, in accordance with Braun and Clarke's six-stage model. Transcripts were coded on NVivo by two raters (SC and UR). To reduce bias, a second rater (rater 2, UR) independently

coded 25% of all interviews using the same method as rater 1 (SC). To enhance validity, following their independent assessments, the raters met to compare, contrast and refine existing codes until agreement was reached on the grouping of main themes and subthemes. Themes and subthemes were then presented and demonstrated by representative quotes. All analyses were reviewed by the primary investigator (PI) and results were presented and discussed with the PI and co-investigators on a fortnightly basis.

## RESULTS
### Phase 1: online survey results
#### Research and related staff survey results
A total of 80 people read the information sheet and provided consent. 52/80 (65%) people completed the initial survey question; 'Thinking about the clinical, health or social care research studies you are currently working on or have worked over the past two years, have you used digital tools to assist in the operation, management or coordination of these studies?' of which eight participants answered that they delegated tool use to colleagues and two participants answered 'no' to having used digital tools; one participant provided reasoning for not using digital tools which were 'stick to what worked in the past' (selected from drop-down menu of answers). These 10 participants were redirected to the end of the survey as the study was primarily interested in exploring experiences of people who had first-hand experience of using digital tools within their research. If participants only had experience of one digital tool, they were asked questions about this specific example. One participant answered questions about one digital example, however, did not provide the name of the digital tool so therefore this participant was removed from analysis as the tool could not be categorised under a research stage (eg, participant recruitment).

A total of 41 participants partially or fully completed the survey (see online supplemental table S4 for participant characteristics). As an introductory question, participants were asked to list all of the digital tools they have used over the past 2 years, 11/41 people completed the survey up until this point. 30 participants continued and provided one digital tool example which they deemed to be (1) the least effective tool and/or (2) the most effective tool, with 19/30 participants answering a number of questions about these tools and completing the survey. Table 2 provides an overview of the most effective and least effective tools by research stage.

Most tools referenced were from ongoing clinical trials, with the largest number of digital tools referenced from data collection (most effective digital tools) and participant recruitment (least effective). Most tools were widely available, as opposed to a bespoke tool. Notably, some tools (eg, social media, Microsoft Teams) were listed as both Most Effective and Least Effective tools, highlighting the variation in experiences when using these platforms.

Online supplemental table S5 provides a breakdown of participants' experiences of using the most and least effective tools they listed, including training requirements and knowledge of costs. Participants selected a range of 1–5 goals for using digital tools, of which the most popular research goal for using digital tool was 'reduce time/ increase efficiency'. The most common users of digital tools referenced by participants were research participants (most effective digital tools) and research delivery (least effective digital tools).

In context of most effective tools, on a scale of 0–10 (0=no more effective than other tools, 10=much more effective than other tools) participants gave a mean rating of 8.1±1.3 (see online supplemental table S6). This result suggests that participants were mostly positive about the tools they deemed to be efficient. In contrast, the least effective tools were rated on a scale of 0–10 (0=no less effective than other tools, 10=much less effective than other tools) with a mean rating of 4.3±2.0 (see online supplemental table S7).

#### R&D survey results
A total of 35 people accessed the survey link and completed the consent form. A total of 25/35 (71.4%) people completed the introductory question 'Has your department used digital tools in the set-up, recruitment and management/monitoring of clinical research? (see online supplemental table S8 for participant characteristics) of which three participants answered 'no' to having used digital tools; three participants provided reasoning for not using digital tools (selected from a drop-down menu of answers). 22/35 (62.9%) participants went onto partially or fully completed the survey.

The largest number of the most effective tool examples, provided by the R&D survey respondents, were for management of clinical studies (n=11). However, the specific tool referenced in this category varied substantially, ie in most cases, each tool was only mentioned by one participant (see table 3). Table 4 provides a breakdown of participants' experiences of using the most and least effective tools they listed, including training requirements and knowledge of costs.

### Phase 2 results: qualitative interviews
A total of eight participants took part in the interviews; five participants had completed the survey for R&D employees, three participants had completed the researchers and related staff survey. Participants were employed by a variety of organisations in the West Midlands area. Six main themes were identified across the transcripts, these are summarised in table 5 along with selected quotes relating to each theme, discussed in detail below. Two case studies of most effective tools have been summarised in online supplemental table S9; case study 1 (management) and case study 2 (recruitment).

**Table 2** Overview of most effective and least effective digital tools examples by research stage

| Research stage | Most effective examples | Least effective examples |
|---|---|---|
| Data collection | Castor Electronic Data Capture (EDC)<br>Electronic completion of questionnaires<br>Electronic database to collect participant data from participating hospitals.<br>MS Teams<br>Online data capture tool<br>Patient Identification Centre<br>REDCap tool to upload patient data<br>Teams/Zoom for participant interviews<br>Text messaging | Trial database<br>Randomisation<br>Database for collected intervention information |
| Intervention delivery | Smart survey<br>Trial intervention delivered to participants via online web portal | Digital intervention as it requires hardware to be used by the patient/research team |
| Other | MS Teams | MS Teams |
| Outcome measures | In-house databases | n/a |
| Participant recruitment | QR code for accessing online survey<br>Social media to recruit participants<br>REDCap e-consent system enables GP practices to remotely consent patients to take part in the trial<br>Twitter page for trial<br>Using an online platform to host consent/demographic forms | Text messaging to invite participants to take part in a survey<br>Facebook and Twitter<br>Asking GP practices to use their SMS systems for inviting patients to questionnaire studies.<br>Health Survey<br>Email—to participants<br>SMS (text message)<br>Social media for recruitment<br>Advertising the study's recruitment advert on reputable charities social media accounts<br>Clinical Practice Research Datalink Interventional Research Services Platform system<br>Reports set up to highlight which participants have dropped out in between expressing interest and being randomised. |
| Research study set up | REDCap<br>WhatsApp<br>MS Teams | Teams<br>Online conferencing tools (dependent on the good connection of all participants)<br>Investigator site file templates distributed via online file sharing service |
| Quality assurance | (none listed) | Datix |
| Stage not provided | Castor EDC for sending out surveys to be completed online<br>Remote research meetings with site collaborators | (none listed) |

GP, general practitioner; n/a, not available.

## Theme 1: definition of digital tools in clinical research

Participants perceived a digital tool used within clinical research as a device that can connect to the internet, or a form of online technology. The main reasoning cited for using a digital tool was to increase efficiency within research (table 5, quotes 1.1–1.2). Digital tools have been adopted across all stages of clinical research from recruitment to data collection, it was noted that tools may differ depending on the research stage and/or role (eg, R&D staff may encounter different digital tools compared with research participants).

## Theme 2: COVID-19 pandemic: validating the benefits of working digitally

It is evident that the COVID-19 pandemic accelerated the use of digital tools within clinical research. It was noted how changes to research methodology needed to be implemented quickly; moving from paper-based methods to digital methods. COVID-19 was perceived as a time to illustrate the benefits of digital tools which helped to alter stakeholder's views who may have been pessimistic towards tools prior to the pandemic. Despite changes in working practices, participants perceived attitudes among staff

**Table 3** Most effective digital tools examples by research stage used by employees working in R&D

| Stage of research | Most effective digital tool example | N |
|---|---|---|
| Set up | Excel trackers | 1 (25%) |
| | Informatics | 1 (25%) |
| | Power BI | 1 (25%) |
| | SharePoint | 1 (25%) |
| | | 4 |
| Recruitment | Social media platforms | 4 (66.7%) |
| | PICS | 1 (16.7%) |
| | Contact for research database | 1 (16.7%) |
| | | 6 (16.7%) |
| Management | Electronic signing of contracts | 1 (9.05%) |
| | Power BI | 1 (9.05%) |
| | REDCap | 1 (9.05%) |
| | TriNetX | 1 (9.05%) |
| | DocuSign | 1 (9.05%) |
| | Audit | 1 (9.05%) |
| | Study tracker | 1 (9.05%) |
| | MS Teams (data collection) | 1 (9.05%) |
| | Online case report form | 1 (9.05%) |
| | Patient trackers | 2 (18.2%) |
| | | 11 |

PICS, patient identification centre; R&D, Research and Development.

and research participants to be mostly positive towards tools. The pandemic served as a trial period, to prove that 'the digital way is working', supporting continued digital tool use and innovation in routine practice. There was a sense of resistance from participants to not revert back to seemingly less efficient non-digital methods (ie, paper questionnaires compared with online).

### Theme 3: perceived benefits and drawbacks of digital tools
*Benefits*
Benefits of tools were considered from two perspectives: research participants and staff using digital tools. For research participants, perceived benefits of using tools included convenience and increased opportunities to take part in research. It was also noted that digital tools can account for individual differences of research participants; for example, some research participants may feel more comfortable contributing to a focus group using the chat function on Microsoft Teams rather than speaking aloud. From the perspective of staff, interview participants considered efficiency (e.g., clinical delivery staff spending less time on admin tasks leading to more time with trial participants), easy connectivity between colleagues (particularly if based in different locations) and ease of offering research materials in different languages as primary benefits of digital tools (see online supplemental table S9, case study 2 for most effective digital tool for recruitment) (table 5, quotes 3.1–3.5).

*Drawbacks*
Some of the drawbacks of digital tools referenced were of a practical nature, for example, technical issues such as poor internet connection or researchers may require additional technical support. However, it was noted that face-to-face research studies may also require additional support. Other drawbacks related to connection, specifically difficulty reading interactions online. Despite the ease of meeting with colleagues online, especially when based in different locations and organisations, there was some concern that meeting colleagues online does not always enable people to strengthen or develop relationships in the same way as meeting face-to-face allows. Participants raised concerns around inequalities and lack of inclusivity relating to digital tool use, it was noted that levels of engagement are currently unclear from different research participant groups when using various digital tools. Digital literacy was identified as a potential risk to excluding certain people with a lack of familiarity with using digital tools, especially if research studies are conducted solely using digital tools (table 5, quotes 3.6–3.9).

### Theme 4: selection of a digital tool and recommendations for future digital tool use
*Selection*
Most participants cited having little choice in the digital tool they used, this was mainly due to the fact that most tool examples were widely used in an organisation (eg, Microsoft Teams) and as a result they did not have the authority to select a tool. Despite this, it was recognised that familiarity with widely used tools was beneficial for ease of use. There was a concern that often staff are not aware of what the most appropriate tool would be for a research task. Some bespoke tools were used; two participants described the process of designing and developing a bespoke digital tool in collaboration with clinical staff (see online supplemental table S9, case study 1 for most effective digital tool for management). The most common way staff discovered about digital tools appears to be through word of mouth (table 5, quotes 4.1–4.2).

*Recommendations for future tool selection*
A number of recommendations were proposed to assist staff in future tool selection. There appears to be no standardised process outlining how to choose a tool, therefore, a checklist database was proposed in which staff could input their digital tool requirements (eg, research stage, user type) and a programme could provide appropriate digital tool options. Further recommendations included a central place (eg, website) to share information about tools, case studies of effective tools (by research stage/stakeholder group) and a local/national network to share digital tool expertise (table 5, quotes 4.3–4.6).

**Table 4**   R&D staff summary of digital tool use

| Question type | Question responses | Set up N (%) | Recruitment N (%) | Management N (%) |
|---|---|---|---|---|
| Total number of most effective examples | n/a | 4 | 6 | 11 |
| How did they hear about digital tool | Developed in house | 1 (25) | 3 (50) | 4 (36.4) |
| | Other | 1 (25) | 2 (33.3) | 2 (18.2) |
| | Recommended by others | 2 (50) | n/a | 2 (18.2) |
| | Saw advertised online/social media | n/a | 1 (16.7) | n/a |
| | Approached by digital tool provider | n/a | n/a | 3 (27.2) |
| | Total responses | 4 | 6 | 11 |
| Training required for digital tool | Yes | 2 (50) | 2 (33.3) | 7 (63.4) |
| | No | 2 (50) | 4 (66.6) | 4 (36.6) |
| | Total responses | 4 | 6 | 11 |
| If yes to training, training length | <1 day | 1 (50) | 1 (50) | 4 (57.1) |
| | 1–5 days | n/a | 1 (50) | n/a |
| | >5 days | 1 (50) | n/a | 3 (42.9) |
| | Total responses | 2 | 2 | 7 |
| Widely used or bespoke digital tool | Widely used | 4 (100) | 5 (83.3) | 8 (72.7) |
| | Novel/bespoke to organisation | n/a | 1 (16.7) | 2 (18.2) |
| | Novel/bespoke to project team | n/a | n/a | 1 (9.1) |
| | Total responses | 4 | 6 | 11 |
| Costs associated with acquiring digital tool | I don't know the answer | 1 (25) | 1 (25) | 2 (25) |
| | No fee (eg, open source) | 1 (25) | 2 (50) | 2 (25) |
| | Periodic recurring fee | 1 (25) | n/a | 2 (25) |
| | Other | 1 (25) | 1 (25) | 2 (25) |
| | Total responses | 4 | 4 | 8 |
| Cost-effectiveness (one highly ineffective to seven highly effective) | 3 | 1 (25) | n/a | n/a |
| | 4 | 1 (25) | n/a | n/a |
| | 5 | 1 (25) | 4 (100) | n/a |
| | 6 | n/a | n/a | 4 (50) |
| | 7 | 1 (25) | n/a | 4 (50) |
| | Total responses | 4 | 4 | 8 |

R&D, Research and Development.

### Theme 5: barriers and overcoming barriers
*Barriers*
Several barriers to digital tool use were referenced including attitudes of staff and research participants; one participant explained that older research participants were concerned about the legitimacy of receiving a survey link via email. A commonly cited barrier was the conflict of carrying out a patient centred job (ie, nurse) while finding time to use digital tools. Resources were also cited as a barrier to digital tool use; resources included budget, physical resources (eg, poor internet connection or limited digital equipment), digital expertise (some organisations had access to specialist digital teams embedded in an R&D department) as well as digital literacy (research participants and staff). Available resources varied across organisations and research participants; appearing to be both a key barrier as well as a key driver for digital tool use (table 5, quotes 5.1–5.5).

*Overcoming barriers*
A number of ways to overcome barriers were proposed by interview participants. Some reported providing research participants with a physical location to visit to complete research activities on a computer (if they did not have access to a computer and/or internet connection). Other participants reported using a hybrid approach of both paper and digital tools to carry out the same research tasks. A hybrid approach was undertaken to reduce digital exclusion, catering for research participants who may not have access to digital tools/internet and/or prefer paper-based methods. However, an impact of this approach included a larger workload due to increased data management.

**Table 5** Summary of identified themes and representative quotes

| Theme | Subtheme | Example statements |
|---|---|---|
| Definition of Digital Tools in Clinical Research | n/a | (1.1) 'Anything that you can use online basically. So, it can be a handset, it can be a tablet or a website…it's anything that can connect to the Internet that can ease the process of undertaking research.'<br>(1.2) 'It helps make it easier to collect data…easier to track where research is (…) And then thinking about the patient facing role of digital research…giving devices to patients or using patients own devices to actually report outcome measures directly to research teams…using technology, whether it's Internet based technology or sort of phone, tablet, laptop-based technology to hopefully improve research even like an e site file system or a trial master file or trial master file system or just using teams and SharePoint, this is a digital solution.' |
| COVID-19 Pandemic: Validating the benefits of working digitally | n/a | (2.1) 'We wanted to continue the research where we could do so…we had to move it online. And people just embraced that. We had our IT team. We had enough resources to get everything onto online platforms in order to collect the data on an online questionnaire as opposed to a paper questionnaire.'<br>(2.2) 'We proved that the digital way is working, so that's massive. It definitely gave us the push needed cause we had no choice. People had to start using digital tools and actually once we got through it (Covid) (…) you go actually, we're not gonna go back to that way before because this way works really well…and it gave us a lot more buy in, particularly with some nurses who may be resistant to change.' |
| Perceived benefits/ drawbacks | 3a. Perceived Benefits | (3.1) 'For the patients that are happy and IT savvy and things it saves a trip to the post box or whatever to fill in the paper questionnaire if people are happy to do it, especially because it's like transferable, you can answer it on your whether it be a mobile phone or a tablet or a PC.'<br>(3.2) 'We potentially can reach patient populations that wouldn't normally take part in research, those that potentially have a lower mobility, aren't able to travel as much could actually take part in the trial.'<br>(3.3) 'You can ask people not only to talk but also to use the chat, which I think helps some people who are bit kind of less able to interact in a group.'<br>(3.4) 'It's easier on an online tool to have a number of different languages …if you're printing off something that's a hell of a lot of paper…and you're sort of sometimes guessing what language people are wanting things in. If there's a drop-down box that could all of a sudden change it to French or you know Arabic, it's a much easier thing to be able to do.' |
| | | (3.5) 'WhatsApp has been used to keep contact with researchers between meetings, which I felt was quite effective…so it helps to have a WhatsApp group and say what are you doing? Can I help? It helps between meetings to keep things going.' |
| | 3b. Drawbacks | (3.6) 'In my experience online is good to get a lot of work done rapidly but sometimes you can't read face to face interactions. You can create maybe bonds that are bit beyond the business like you can have a coffee together so there's a little bit of a difference there.'<br>(3.7) 'Some participants might struggle with connection and so on. So, there's a number of technical issues that [could occur] but again that's very similar to a logistical issue that you would have in a face-to-face workshop.'<br>(3.8) 'The further negatives are probably people having to use a new system if they don't use it that often then potentially forgetting how to use it. But obviously we've tried to … reduce that by offering videos, offering…training sessions.'<br>(3.9) 'I think drawbacks are still engagement and I think this is what we're trying to investigate at the moment… what is engagement like across different participant groups and age ranges and demographics and ethnicities and stuff like that… I'm not sure it is inclusive.' |
| Selection of a Digital Tool and Recommendations for Future Digital Tool Use | 4a. Selection | (4.1) 'It was basically chosen because it was free, and that's what the university had'.<br>(4.2) 'Things tend to be sort of by word of mouth I think a lot relies upon the fact that you happen to have a conversation you know when somebody says ohh, hold on a minute you could do it this way or you could use this.' |
| | 4b. Recommendations | (4.3) 'Almost a digital checklist and then it points to what program, even if it doesn't say what the program is, it tells you the type of program you need…depending on where you are with it and what you're doing. Are you the sponsor? Well, if you sponsor a study you'll need a very different [tool] to if you're delivering the study… is it multi-site?.'<br>(4.4) 'I think it would be good to have case studies of people who have used some tools and made their own adaptations [for a] service user or participant group.'<br>(4.5) 'If you don't know what's available you, you don't know what you don't know. There could be a digital solution to the problem you've got but if that's not your background, if you haven't encountered it, you wouldn't know to search for it…. I think a checklist…suggestions or case studies essentially and almost you know sharing best practice across regions…It could be national.'<br>(4.6) ' A community of practice might be helpful because then you can bring people together (…) and (…) exchange information. So, I guess developing a network of practitioners working [on] research using tools… that might help to improve the way we use them.' |

Continued

**Table 5** Continued

| Theme | Subtheme | Example statements |
|---|---|---|
| Barriers and Overcoming Barriers | 5a. Barriers | (5.1) 'It feels like people were happy to make the move because you couldn't [use] paper. But following COVID some people have gone back to paper…especially older people (…) they said…that they'd be worried about clicking on a link and whether it was, you know, bonafide or not…whether it is safe to click on the link.'<br>(5.2) 'Digital tools are there for everyone. Some teams grab with them and run with them really quickly. Others are really resistant still and I think it's what you know. And it's not clear cut on age at all, but there definitely is a demographic that can feel more comfortable with it, and others that don't and I think that comes down to experience.'<br>(5.3) 'They are trained as nurses first and foremost (…) their first priority is always gonna be their patients and you know, hear stories about them working super long shifts, they don't take a break for the entire time, when are they gonna have time to sit down and play with Excel.'<br>(5.4) 'If they've got no budget for IT, then they'll need to whip up an Excel spreadsheet.'<br>(5.5) 'We've got ideas as to what would make life easier for us, which ultimately makes life easier for staff and for patients (…) we don't have the expertise behind it so we would be expecting other people to do it. So, the barrier really is sometimes the experience of putting your idea into something that will actually work. This cost is always going to be a barrier in the NHS…if we don't invest in something, how are we ever going to move forward… and so if we invest in there, then actually we might be able to start, you know, save on staff time. So, I think the biggest barrier is expertise.' |
| | 5b. Overcoming Barriers | (5.6) 'I did a questionnaire study and used 18 practices…I sent a pre-notification text to nine practises and nine practices I just sent the link to the study and there was absolutely no difference in participation rate.' |
| | | (5.7) 'We keep an open mind as to whether people want to show their face or not. Because some people are not comfortable with showing their house, you know it's quite invasive in a way (…) if some people want to keep the cameras on, we say we prefer them to keep the camera on. But if they want to keep off and just participate by audio (we're) also quite open to that.' |
| | | (5.8) 'There's a potential that maybe people who aren't technologically aware would struggle to take part in the study (…) it isn't an all or nothing because this is a remote study it has to be remote, no you can actually take that remote nature away and just have a patient complete it on a practice computer.' |
| Future Digital Tool Use | n/a | (6.1) 'Maybe in 20 years and everybody will really use them. So digital exclusion might reduce over time I feel.'<br>(6.2) 'Clinical trials seem to be becoming more complex (…) I think people are more inclined to look at digital support with studies (…) I think historically (…) nurses would be expected to pick up these sorts of administrative tasks. I think now we're looking at getting people in who can do it and it's a lot more efficient.'<br>(6.3) 'Researchers here come and say to me…what's your evidence for using it? What's are your response rates for those via paper and those that answered via e-mail or text message? (…) I just think that needs to be published a bit more so people can make informed decisions around what they're going to use for their particular sample.' |

It is evident that individual differences exist with digital tool use, with interview participants described adopting a flexible approach with research participants. For example, some research participants may not feel comfortable using their video camera when using Microsoft Teams. Interviewees explained how research participants and staff attitudes can be a barrier to digital tool use but offering training in multiple modalities can help to increase users' confidence (eg, video, face-to-face demos). Some interviewees mentioned conducting their own research to evaluate the effectiveness of digital tools. For example, one participant had previously compared two recruitment methods; a text message in advance informing participants they would receive a survey link via email compared with no prenotification at all. However, this study reported no differences in the number of participants recruited to the study via either method, suggesting that reassurance of the legitimacy of the survey link did not have any impact in terms of increasing engagement (table 5, quotes 5.6–5.8).

### Theme 6: future digital tool use

All participants perceived they would be using tools increasingly in the future and that digital exclusion of research participants may reduce over time due to increasing familiarity with digital tools. Participants highlighted the potential for pressure to be placed on nurses to keep up to date with digital tools while caring for patients, as well as the need for collaboration between digital experts and clinical staff when developing and utilising tools. Suggestions for research to inform future tool use included investigating which tools are most effective for different research stages and stakeholder/participant groups and research exploring the cost-effectiveness

of digital versus paper-based tools. Further suggestions included organisations conducting training analysis of all stakeholders using digital tools (table 5, quotes 6.1–6.3).

## DISCUSSION

This mixed-method study aimed to explore which tools are being used across clinical research as well as staff experiences of digital tool use. Our findings indicate that digital tools are now a fundamental part of clinical research; the COVID-19 pandemic accelerated the use of tools with adoption observed across all research stages.[1 11–13] Tools were mostly viewed favourably, with increased efficiency cited as a leading benefit for research staff. It was noted that tools can account for individual differences of research participants, potentially offering research opportunities to a broader sample of participants.[1 8 14 15] For example, materials can be easily offered to research participants in different languages and reduce participants' need to travel to a physical location. Despite this, some drawbacks and barriers to the use of tools exist. In particular, there were concerns about digital exclusion of research participants who may not have access to digital tools, and/or internet connections, or lack familiarity with using these platforms.[1 16–18] There was also a concern about limited resources within clinical research; particularly, the need for individuals with digital expertise to advise on and develop digital tools.

A previous mixed-methods study by Blatch-Jones et al[8] explored which digital tools were being used for recruitment and retention within clinical trials. The current study extends this research by investigating UK stakeholders digital tool use in the context of the COVID-19 pandemic as well as across a wider range of research stages (ie, informed consent, management, data collection), considering not only the benefits, but also the limitations of digital tools. Consistent with the afore mentioned study by Blatch-Jones et al,[8] our findings add to existing evidence that digital tools provide opportunities to widen reach of participation as well as reflecting previous concerns around digital exclusion for some groups.[1 8 19 20] New insights from this study compared with Blatch-Jones et al[8] include suggestions for overcoming barriers to digital tool use. Recommendations included attempts to reduce digital exclusion by providing training on digital tools in various modalities (eg, videos, face to face) to increase research participant's/staff confidence in use and providing a physical location and/or digital tool in order for people to participate in a study. Further insights included issues around digital tool selection, participants proposed the need for a standardised process for selecting the most appropriate tool as well as published case studies of effective tools. Indeed, it has been previously reported that staff are often not sure of tool appropriateness for a particular task, research stage or stakeholder group, and therefore, a formalised selection process is required.[1 8] The need for collaboration with clinical staff and digital expertise was also highlighted in the context of developing digital tools in the future, case study 1 (online supplemental table S9) provided an example of a cost-effective bespoke tool.

Consistent with previous research, our study supports concerns around the security of data when using digital tools, particularly among older age research participants and invitations to participate via text/email.[14 21] It is important that research participants perceive invitations as authentic and legitimate otherwise this may negatively affect study uptake.[8] In this study, a participant reported conducting research to compare the recruitment rates of a prenotification text to inform participants they would be shortly receiving a survey link compared with no text. However, no difference was found between the methods on participant uptake. This indicates such an approach may not be effective in improving the credibility of the subsequent message. It is critical that participants can trust recruitment invitations for clinical research, with the opportunity for scam messages being minimised, based on the way invitations are distributed.

This study highlights several considerations for research staff and organisations when using digital tools for research. Research projects should aim to be inclusive for people who may not have access to digital tools or the internet (eg, providing a physical location/digital tool for people to complete the study).[1 19] Organisations need to consider whether adequate training is provided for both staff and research participants to help address gaps in knowledge, demystify tools and increase confidence.[1 22] It is apparent that one size does not fit all, research staff should be willing to accommodate personal preferences of research participants when using digital tools (eg, option to keep video camera off on video conferencing platforms) as well as consider offering a hybrid approach (both paper and digital tool options). In addition, staff should capitalise on digital expertise both locally in their organisations and nationally via funders and other relevant organisations.

Strengths of this study include the use of a broad range of stakeholders as participants and the use of mixed methods, enabling exploration of digital tool use from multiple perspectives. However, a limitation of this approach is the wide scope of the survey questions (ie, covering all research stages) which does not allow deeper exploration into specific tools. It was apparent that many of the tools identified within the surveys were only referenced once by a single participant, suggesting a possible lack of consistency in the tools used within clinical research and the need for standardisation across each research stage. Alternatively, another explanation could be the small number of survey respondents which could make it difficult to generalise the results. Future research could elaborate on this work and investigate tools used in different geographical regions. A further limitation of this study is not exploring the views of participants/carers due to limited participant response. Given that our findings, alongside previous research, suggest that participants/carers are a key user of digital tools, future research should seek to understand their experiences, ensuring there are no barriers to accessing/using digital tools to prevent digital exclusion and promote diversity in participants. It is likely that participants were motivated to participate due to an underlying interest in digital tools, consequently some perspectives may

not have been included in this study (staff, participants/carers who have not used tools).

It is evident that future research should develop a number of resources to help researchers choose appropriate digital tools. Our research findings alongside previous studies have recognised requirements to publish case studies of effective digital tools (specific to research stage/stakeholders) as well as develop a standardised process of choosing a tool and/or programme for recommending appropriate tools.[15] A target for the funders and organisations related to clinical research is to develop local and national networks to share and showcase expertise in digital tools. Further research avenues should include an evaluation of the effectiveness of tools (specific to the research stage) and an economic analysis to better understand the cost-effectiveness of digital tools (particularly in relation to non-digital approaches) which would help to bolster the argument for use of digital tools in clinical research.

**Acknowledgements** The authors thank all study respondents taking part in the study. The authors also thank the steering group for their contributions as well as the participants who provided case study examples of effective digital tools.

**Contributors** CC, TNA, CT and MElliott obtained funding and conceptualised the research. MElliott and SC drafted initial online survey questions and qualitative interview questions; SC, CC, TNA, MElliott, MEvans and CT, reviewed the draft survey/qualitative questions. SC, MElliott, CC and MEvans were directly involved in activities relating to participant recruitment. SC and UR tidied the qualitative interview transcripts and analysed qualitative data. SC and MElliott conducted data analysis for online survey results and drafted the manuscript, all authors reviewed the final manuscript. MElliott is acting as the guarantor for this study.

**Funding** This study was supported by the National Institute for Health and Care Research, Clinical Research Network West Midlands Improvement and Innovation Strategic Funding.

**Competing interests** CC and MEvans are employed by the funder (NIHR) of this project. The review is part of a wider project which CC helped to conceptualise. CC and MEvans have revised and helped finalise the online survey and interview questions while also providing guidance about digital tools used in clinical research. CC and MEvans were directly involved in activities relating to participant recruitment.

**Patient and public involvement** Patients and/or the public were not involved in the design, or conduct, or reporting, or dissemination plans of this research.

**Patient consent for publication** Not applicable.

**Ethics approval** This study involves human participants and was approved by the University of Warwick's Biomedical and Scientific Research Ethics Sub-Committee (111/20-21). Participants gave informed consent to participate in the study before taking part.

**Provenance and peer review** Not commissioned; externally peer reviewed.

**Data availability statement** All data relevant to the study are included in the article or uploaded as online supplemental information. Please see online supplemental files for all additional data.

**ORCID iDs**
Sophie Clohessy http://orcid.org/0000-0003-2945-157X
Theodoros N Arvanitis http://orcid.org/0000-0001-5473-135X
Carla T Toro http://orcid.org/0000-0001-6351-1340
Mark T Elliott http://orcid.org/0000-0003-4000-0198

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
