## [Reviewer comments · BMJ Open]

ARTICLE DETAILS

TITLE (PROVISIONAL)	Using Digital Tools in Clinical, Health and Social Care Research: A Mixed-Methods Study of UK Stakeholders
AUTHORS	Clohessy, Sophie; Arvanitis, Theodoros; Rashid, Umer; Craddock, Carly; Evans, Mark; Toro, Carla; Elliott, Mark

VERSION 1 – REVIEW

REVIEWER	Moradian, Saeed York University Faculty of Health, School of Nursing
REVIEW RETURNED	06-Jul-2023

GENERAL COMMENTS	The article is well-structured and provides valuable insights into the use of digital tools in clinical research. To further improve the article, here are some general comments and feedback: Abstract: Some sentences in the abstract could be rephrased for clarity and readability. For example, the sentence "Tools were mostly perceived in a positive light, offering increased efficiency for research" could be revised as "The majority of participants viewed the tools positively, noting their ability to enhance research efficiency." Introduction: • While the authors mention the need to explore experiences of digital tools across different research stages, it could be beneficial to explicitly highlight the existing gap in the literature. Specify how previous studies have primarily focused on recruitment and retention, leaving other stages understudied. This will help emphasize the novelty and contribution of your research.• Since the study specifically aims to investigate the impact of the pandemic on the switch to digital tools, it would be helpful to provide more information on the challenges faced by researchers during this transition. Elaborate on how the pandemic necessitated the rapid adoption of digital methods and why understanding these experiences is important.• The research objectives could be revised to provide more clarity. Instead of listing various aspects of clinical research (e.g., setup, management, recruitment), consider categorizing them under broader themes (e.g., research operations, participant engagement). This will provide a clearer framework for the study. Methods: 1. Phase 1: Online Surveys • Specify the total number of participants targeted for each survey type (Researchers and Related Staff, Research & Development Staff) to provide a clearer understanding of the sample size.
--

	 • Clarify the method of recruitment for participants in each survey type (e.g., invitation emails, online platforms) to provide transparency on how participants were selected. Also, add the approximate number of potential participants within each participant group to provide context for the target population size. 3. Table 1: Please reformat the table to improve readability and ensure clear differentiation between the survey types, eligibility criteria, and participant groups. Recruitment:  1. Instead of referring to it as "a database of clinical, health and social care research studies," provide the specific name or title of the database, such as the National Institute of Health Research Clinical Research Network (NIHR CRN) West Midlands database. 2. Please clarify the selection criteria: In addition to mentioning that projects within the past three years were identified, provide more details on the specific criteria used to select projects for inclusion in the survey. This will help readers understand how the projects were chosen and ensure transparency. 3. Please provide reasoning for the second phase of recruitment and explain why the decision was made to conduct a second phase of recruitment. Was it due to low response rates or the need to increase participant numbers? Procedure:  1. Please provide a brief description of the Qualtrics platform and its purpose. This will help establish the context for participants' interaction with the surveys. 2. Please enhance information about consent and debriefing e.g., instead of mentioning the consent form and debriefing sheet separately, consider combining the information into a single sentence or paragraph. Also, provide a clear transition between the two surveys. Data Analysis:  1. Please specify the types of descriptive statistics used e.g., mention whether you used measures such as means, frequencies, percentages, or other relevant statistical measures to analyze the multiple-choice and scale answers. This will provide a clearer understanding of how the data were summarized. 2. Please provide more information about the thematic analysis approach used. Also, explain how themes were identified, coded, and organized from the qualitative data. 3. Please provide details on how the results were verified with the project team. Explain the process of reviewing and validating the findings to ensure accuracy and consensus. This will demonstrate the rigor of the analysis process and the efforts made to ensure the trustworthiness of the results. Results: Phase 1:  1. Clarify the reasons for excluding participants: When explaining the exclusion of certain participants, provide more details on the criteria used for exclusion. For example, clarify why participants who delegated tool use, did not use digital tools or did not provide the name of a digital tool were redirected to the end of the survey or removed from the analysis. This will help readers understand the rationale behind these decisions. 2. Enhance the presentation of participant completion rates e.g., instead of simply stating that 41 participants partially or fully completed the survey, provide the completion rate as a percentage of the total participants invited. 3. In the paragraph mentioning participant characteristics, consider including additional details e.g., age, gender, or professional roles
--	---

	to provide a more comprehensive profile of the participants. This will add depth to the understanding of the sample. 4. Table 2&3&4 can be better formatted e.g., properly labeled with column headings. For questions in Table 4 that have a yes/no response or multiple-choice options, consider providing the percentages alongside the counts. Also, instead of solely presenting the data in tables, offer some insights or analysis of the findings. Highlight any patterns, trends, or notable points that emerged from the responses. This will add depth to the results and help readers better comprehend the implications of the findings. Results phase II:  1. Table 5, it would be beneficial to number the themes and sub-themes for ease of reference throughout the section. 2. It seems that there are no specific sub-themes mentioned under Theme 1 ("Definition of Digital Tools in Clinical Research"). If there are indeed sub-themes, it would be helpful to list them in the table and provide a brief explanation of each sub-theme in the text. 3. Theme 6 is listed as "N/A" in the table, but some statements related to future digital tool use are mentioned in the text. To maintain consistency, either include the relevant statements under Theme 6 in the table or adjust the table accordingly if this theme is not relevant to the qualitative interviews. Discussion:  1. It would be helpful to provide specific references for the studies mentioned in the discussion (e.g., Barakat et al., 2021; Everhart et al., 2021; Fisher et al., 2021; Masoli et al., 2021; etc.). This will allow readers to easily locate and access those studies if they are interested in exploring the topic further. 2. Please clarify the link between findings and previous research e.g., when discussing the findings in relation to previous research, it would be beneficial to explicitly state how the current study's findings align or differ from previous research. This will help to establish the contribution and novelty of the current study in the field. 3. Expand on the limitations: While the discussion briefly mentions the limitation of not exploring the views of participants/carers, it would be valuable to discuss the potential impact of this limitation on the overall findings and the implications for future research. Additionally, any other limitations of the study should be acknowledged and their potential impact on the interpretation of the results should be discussed.
--	---

REVIEWER	Jeffries, Mark University of Manchester, School of Health Sciences
REVIEW RETURNED	11-Jul-2023

GENERAL COMMENTS	Using Digital Tools in Clinical Health and Social Care Research: A Qualitative Study of UK Stakeholders. Thank you for the opportunity to review this paper which I found interesting to read. This covers an important area of concern for researchers and should be of interest to many involved in health and clinical research. I have a few points for consideration by the authors in revising the manuscript. I hope they are of some help. Title and general point  1) I think the title is inaccurate considering that half the paper is survey data, some of which quantified and presented as numbers and percentages. I would suggest that this should be a mixed methods study.
---

	2) I find the scope of the paper very broad, covering many different types of digital technology which possibly makes specific and tailored recommendations difficult. There is also a huge difference between the use of large datasets for cross sectional epidemiological studies, digital technologies for clinical trials and digital technologies used in qualitative data collection. I'm unsure whether the differences and distinctions between these are brought out in the paper. Essentially there's not much that can be done about the breadth, since that's what you set out to find in some ways, and what your study was about, so I'm not suggesting a narrowing of focus, but I do think that scope and breadth needs greater acknowledgment and discussion both in the introduction and discussion sections. Abstract 3) Could you please explain the reasoning behind the decisions to boundary recruitment to one geographical area and to the last two years? Strengths and Limitations bullet points 4) You highlight as a strength that the survey and qualitative questions were co-designed with an experienced stakeholder group. I can't find, and I may have just missed, this mentioned in detail in the body of the manuscript. Could you please provide details of who this stakeholder group was and details of how the questions for the survey and interviews were developed. Introduction 5) I feel the introduction needs more detail and more background including referencing a greater range of previous literature. There's somewhat a lack of clarity here. You mention, in parentheses, a number of different digital tools and do the same when discussing how these are used in different stages of research. In the second paragraph on page 5 you don't mention data collection tools but then do in the aims of the study just below. Could you provide, maybe in a table, as a suggestion, the sorts of digital tools used and at the various stages in research they are used. I feel this section requires greater clarity to enable the reader to fully understand why this is important and how specifically, the roles of different digital tools in research have been previously understood. 6) Related to the above regarding the different research stages in which digital tools can be used you mention only one previous piece of research here relating to recruitment and retention but nothing relating to other stages of research. Other research about other stages might help – for instance Shamsuddin et al on Online Workshops during Covid 19 might be useful (I am not an author of that paper so have no stake!) I think adding further literature here would be useful in contextualising how your study aimed to build upon that. Methods 7) Generally I feel the methods are really detailed and clear. 8) Consent for the surveys – was it embedded in the Qualtrics platform or sent as an additional document? 9) Page 7 line 14 – Typo missing apostrophe “participants' experience” 10) Page 7 Lines 15-16 section in brackets – not clear what you mean - please expand. 11) Data saturation. I agree with your definition here but exactly how did you decide no further themes could be interpreted? 12) If you are going to put Thematic Analysis in capitals you might need to cite Braun and Clarke 13) Great detail in the analysis section but maybe this also needs something on how you moved from the final coding to the
--	---

	interpretation of themes. Feels that there might be a step missing here. Results 14) Overall, I think these are really interesting, with valid and interesting themes and they read really well. 15) 19/52 completed all the first survey and 22/25 the second. Can you please, somewhere, reflect upon these numbers and whether you felt they were sufficient for the interpretations you make? 16) Similarly, as I understand, and I may have this wrong, each digital tool is mostly only mentioned once. Does that tell us much about their use? 17) Table 5 is really useful but I feel that most qualitative studies also embed some quotations in the text. It is somewhat difficult to jump back and forward from the text in the results to the table. Discussion 18) Para 2 Page 23. This relies quite heavily on one previous study. Is there anything else from previous literature that your findings build upon? 19) Your findings and considerations around inclusivity are really interesting and important. References 19) Related to some of my points above. You draw upon and cite only 14 references. I would have expected a study such as this with this sort of scope to draw upon a wider literature. PPI 20) You had no PPI involvement. Could you reflect on why this was the case and perhaps discuss this as a limitation?
--	---

REVIEWER	Wyatt, David King's College London, Division of Health & Social Care
REVIEW RETURNED	15-Jul-2023

GENERAL COMMENTS	Thank you for giving me the opportunity to review this interesting paper, which draws on surveys and semi-structured interviews to explore which digital tools are used in clinical research, the experiences of a broad range of staff of these tools and their effectiveness. The authors present some rich data (particularly in Table 5), and I think this paper has a lot of potential and is of value to readers of BMJ Open. In its current form, it has some flaws and I hope the comments I provide below help the authors to refine this paper for publication. Introduction At the moment, the reader is not provided with any definition or understanding of how the authors are operationalising 'digital tools' in this paper. I think this needs to be added (even if just using the definition provided to survey participants in Table S1). The introduction also mentions the effectiveness of tools as one of the aims of the study (absent from the Strengths and limitations bullets or the paper's Discussion. I can see that this was covered in the survey data which is currently in a supplementary table. If effectiveness is going to remain, it needs to be discussed in some detail in the paper itself (although it might be easier just to remove effectiveness as there's already a lot here to cover). Methods
--

	While the methods section is detailed, there is not a single reference to methods literature. Surveys - In the Strengths and weaknesses bullet points, the authors mention that the surveys were codesigned with an experienced stakeholder group. A little detail about this process would be really valuable here. This is particularly important because later on page 8 the authors mention a selection from a dropdown list of reasons for not using digital (which I assume was part of the co-design process). 8 interviews is a very small number to claim data saturation, particularly when the number of digital tools available and their potential uses are so broad, and you had two very large staffing groups (R&D staff (n 5)- there are so many roles here that link to research, and Researcher and related staff (n3) - from CRNs, CRPs, university research staff, trial managers etc. My gut feeling is that it might be better to talk about these interviews as a means of adding further texture to your survey data (which is what I understood was the purpose of them anyway) and leave it there. If the authors want to make this claim of data saturation, it would be good to add a sentence or two explaining how exactly they decided that saturation was met (so that this claim is a little more robust). For example, I assume you were analysing as you went along, so you might have done 6 interviews, realised you were getting similar things, and then been more aware of new points (or the lack thereof) in the last 2 interviews. The team might have looked at the data and agreed no new codes were identifiable. Results The survey results at the moment have a short paragraph and a table per survey. There is a lot of data here to process, and some rather small numbers. It would be good to add one or two more paragraphs signposting the reader to particular pieces of data you want them to focus on. Are the authors able to give any sense of how many people each of the surveys were sent to/the response rates? The numbers are quite small so any context here would be useful to know. Thematic analysis of qualitative data I struggle a little with this thematic analysis because the themes match so neatly on the question topics listed in Table S3. As such they're really quite descriptive rather than analytic. While these broad topics/headings seem useful in structuring the data, I think a deeper look at some of the points within it would be useful. For instance, on page 17 theme 2 "impacts of the Covid-19 pandemic", the last sentence is the most interesting point here (in my opinion) - not that covid-19 accelerated the use of digital (this has been said before). But that covid provided a way to, as Table 5, point 2.2 says, "proved that the digital way is working" so that it remains in routine practice and continues to be a space of innovation. Could this point be developed more? (if appropriate)). Table 5, Theme 3, Drawbacks - an interesting theme that appears for me here is one of connection, both in the human and the WIFI
--	--

	sense. Another theme is of inequalities. I obviously am only seeing a snapshot of the data in Table 5, but it seems that revisiting the data and thinking beyond the existing themes and more analytically would be useful. Discussion The discussion summarises the paper well and presents some implications for the future. I think some of this text will need to be updated if the authors revisit the analysis.
--	--

VERSION 1 – AUTHOR RESPONSE

Reviewer 1 (from PDF):

The article is well-structured and provides valuable insights into the use of digital tools in clinical research. To further improve the article, here are some general comments and feedback:

Abstract:

2.1. Some sentences in the abstract could be rephrased for clarity and readability. For example, the sentence

"Tools were mostly perceived in a positive light, offering increased efficiency for research" could be revised as "The majority of participants viewed the tools positively, noting their ability to enhance research efficiency."

This sentence has now been changed for clarity to the following: "The majority of participants viewed the tools positively, noting their ability to enhance research efficiency" (page 2).

Introduction:

2.2. While the authors mention the need to explore experiences of digital tools across different research stages, it could be beneficial to explicitly highlight the existing gap in the literature. Specify how previous studies have primarily focused on recruitment and retention, leaving other stages understudied. This will help emphasize the novelty and contribution of your research.

This has now been amended on page 4.

2.3. Since the study specifically aims to investigate the impact of the pandemic on the switch to digital tools, it would be helpful to provide more information on the challenges faced by researchers during this transition. Elaborate on how the pandemic necessitated the rapid adoption of digital methods and why understanding these experiences is important.

Further information has now been added on page 3. "The COVID-19 pandemic accelerated changes to clinical research methodology, social distancing forced study teams to transition from conducting research activities in person to predominantly online and/or via remote means (e.g., 1, 5, 6). Documented challenges associated with rapid adoption of digital tools in clinical research include;

staff/participant lack of familiarity with or limited access to digital tools and rapid changes to protocols/ethics applications to ensure compliance (5)”

2.4. The research objectives could be revised to provide more clarity. Instead of listing various aspects of clinical research (e.g., setup, management, recruitment), consider categorizing them under broader themes (e.g., research operations, participant engagement). This will provide a clearer framework for the study.

Thank you for this suggestion. However, to maintain consistency throughout we preferred to keep the objectives as they were to align with the design of the online surveys and interview guides.

Methods:

Phase 1: Online Surveys

2.5. Specify the total number of participants targeted for each survey type (Researchers and Related Staff, Research & Development Staff) to provide a clearer understanding of the sample size.

This has been added to page 6.

2.6. Clarify the method of recruitment for participants in each survey type (e.g., invitation emails, online platforms) to provide transparency on how participants were selected. Also, add the approximate number of potential participants within each participant group to provide context for the target population size.

Under the recruitment section on page 5, sub-headings have now been added for each survey. Further detail has also been added regarding each recruitment method and recruitment numbers on page 6.

2.7. Table 1: Please reformat the table to improve readability and ensure clear differentiation between the survey types, eligibility criteria, and participant groups.

Table 1 has now been amended, an additional column for ‘participant group’ has been added, see below.

Table 1: Eligibility criteria by survey type

Survey Type	Participant Group	Eligibility Criteria
Researchers and Related Staff	People working on clinical research projects either in a research capacity or operational support.	Aged 18 years or older; employed by an organisation based in the West Midlands region of the UK; working on clinical research projects over the past two years (2020-2022)

Research & Development (R&D) Staff	People working within a hospital's Research & Development (or similar) department, with a role to support or coordinate clinical research taking place within, or in partnership with the Hospital.	Aged 18 years or older; hospital must be a research active NHS Trust in the West Midlands region of the UK; working in a hospital's Research & Development (or similar) department over the past two years (2020-2022)
---	--

Recruitment:

2.8. Instead of referring to it as "a database of clinical, health and social care research studies," provide the specific name or title of the database, such as the National Institute of Health Research Clinical Research Network (NIHR CRN) West Midlands database.

Further information has been added to page 5-6;

- i) Researchers and Related Staff Survey. Participants were recruited primarily using databases of clinical, health and social care research studies held by the National Institute of Health Research Clinical Research Network¹ - NIHR CRN West Midlands. An initial list of relevant projects were obtained by NIHR staff using the Open Data Platform (ODP) that transforms the data held in the NIHR's Central Portfolio Management System (CPMS) into a usable form, allowing it to be tabled, filtered, and organised by various metrics. Projects were selected based on the following criteria: Clinical, health or social care projects led by the Clinical Research Network West Midlands, that had been completed or were ongoing within the previous three years² (with any recruitment activity during the four financial years 2019/20 - 2022/23, filtered to include only those studies opened since 01/06/2019). Additional data was sourced from CPMS and EDGE, the Local Portfolio Management System (LPMS), which functions as a secure data storage platform for the records of study approvals, delivery & site activity at a regional level. An email invitation and survey link were then distributed to the lead researcher or main contact for the selected research studies.
- ii) R&D survey. A total of 26 NHS Trust R&D departments in the West Midlands were contacted by NIHR staff via email invitation and survey link.

2.9. Please clarify the selection criteria: In addition to mentioning that projects within the past three years were identified, provide more details on the specific criteria used to select projects for inclusion in the survey. This will help readers understand how the projects were chosen and ensure transparency.

Please see response to comment 2.8 above.

2.10. Please provide reasoning for the second phase of recruitment and explain why the decision was made to conduct a second phase of recruitment. Was it due to low response rates or the need to increase participant numbers?

A second phase of recruitment was conducted due to low response rates, explained in the text on page 6; We anticipated approximately 50 people to complete each survey, however after the first data collection period (27th June - 26th July 2022), participant numbers were lower than expected (35 for Research and related staff survey and 15 for R&D); therefore, a second phase of recruitment took

place (14th October - 15th November 2022). An email prompt was sent to all contacts via the same mechanism as the initial survey invitation.

Procedure:

2.11. Please provide a brief description of the Qualtrics platform and its purpose. This will help establish the context for participants' interaction with the surveys.

This has now been added to page 6. "Participants accessed surveys via a hyperlink. The surveys were created and hosted on Qualtrics, a secure online survey platform (<https://www.qualtrics.com/uk/>), which is used for creating surveys with in-depth response formats and allows for sophisticated methods of distribution and data management (10)".

2.12. Please enhance information about consent and debriefing e.g., instead of mentioning the consent form and debriefing sheet separately, consider combining the information into a single sentence or paragraph. Also, provide a clear transition between the two surveys.

Information has been combined for consent and debriefing on page 7. "Before taking part, all participants read an information sheet and ticked a consent form (embedded into the Qualtrics platform) to agree to their participation, after survey completion, participants were presented with a debriefing sheet which repeated the objectives of the study and information about how the survey data would be stored".

To provide a clear transition between the two surveys, sub headings for each survey procedure has been enhanced, on page 7; "Researchers and Related Staff Survey: Procedure Overview" and "R&D Staff Survey: Procedure Overview".

Data Analysis:

2.13. Please specify the types of descriptive statistics used e.g., mention whether you used measures such as means, frequencies, percentages, or other relevant statistical measures to analyze the multiple-choice and scale answers. This will provide a clearer understanding of how the data were summarized.

This information has now been added to page 8; "Multiple choice and scale answers were analysed using frequencies and percentages".

2.14. Please provide more information about the thematic analysis approach used. Also, explain how themes were identified, coded, and organized from the qualitative data.

Further information has been added to the data analysis section on page 8-9;

Transcripts were anonymised and cross-checked for accuracy by the first author (SC) and Research Assistant (UR) against the recordings. Interview transcripts were then uploaded into NVivo qualitative analysis software (Version 12, QSR International, Massachusetts, US). A Thematic Analysis (Braun & Clarke, 2006) was applied, using an inductive approach, in accordance with Braun and Clarke's six-stage model. Transcripts were coded on NVivo by two raters (SC and UR). To reduce bias, a second

rater (rater 2, UR) independently coded 25% of all interviews using the same method as rater 1 (SC). To enhance validity, following their independent assessments, the raters met to compare, contrast, and refine existing codes until agreement was reached on the grouping of main themes and subthemes. Themes and subthemes were then presented and demonstrated by representative quotes. All analyses were reviewed by the Primary Investigator and results were presented and discussed with the PI and Co-Investigators on a fortnightly basis.

2.15. Please provide details on how the results were verified with the project team. Explain the process of reviewing and validating the findings to ensure accuracy and consensus. This will demonstrate the rigor of the analysis process and the efforts made to ensure the trustworthiness of the results.

Further information has been added to page 8-9.

Results:

Phase 1:

2.16. Clarify the reasons for excluding participants: When explaining the exclusion of certain participants, provide more details on the criteria used for exclusion. For example, clarify why participants who delegated tool use, did not use digital tools or did not provide the name of a digital tool were redirected to the end of the survey or removed from the analysis. This will help readers understand the rationale behind these decisions.

Further information has been added to help the reader understand the rationale behind the exclusion decisions (page 9).

2.17. Enhance the presentation of participant completion rates e.g., instead of simply stating that 41 participants partially or fully completed the survey, provide the completion rate as a percentage of the total participants invited.

This information has been added for the Researcher and Related staff on page 9; for the R&D survey on page 12. We have included the number of participants who read the information sheet and consent form, followed by the number of participants who answered initial questions and then the survey itself. It is difficult to estimate the number of participants invited as the study was advertised on social media. Further information about survey distribution estimates have been added on page 6.

2.18. In the paragraph mentioning participant characteristics, consider including additional details e.g., age, gender, or professional roles to provide a more comprehensive profile of the participants. This will add depth to the understanding of the sample.

For both surveys, this information can be found in the Supplementary tables, S4 and S8.

2.19. Table 2&3&4 can be better formatted e.g., properly labelled with column headings. For questions in Table 4 that have a yes/no response or multiple-choice options, consider providing the percentages alongside the counts. Also, instead of solely presenting the data in tables, offer some insights or analysis of the findings. Highlight any patterns, trends, or notable points that emerged

from the responses. This will add depth to the results and help readers better comprehend the implications of the findings.

We have followed the journal guidelines for table formatting. Due to the limited word count restrictions from the journal, we wanted to focus the detailed analyses on other aspects of the study (i.e. qualitative results) and include these tables as indicative summary statistics. Highlighted key findings can be found on page 11 and 12.

Results phase II:

2.20. Table 5, it would be beneficial to number the themes and sub-themes for ease of reference throughout the section.

Thank you for this suggestion - Sub themes have now been labelled using letters. Letters have been chosen to avoid confusion between quotes and sub themes for example a sub theme is 3a, but quote is 3.1.

2.21. It seems that there are no specific sub-themes mentioned under Theme 1 ("Definition of Digital Tools in Clinical Research"). If there are indeed sub-themes, it would be helpful to list them in the table and provide a brief explanation of each sub-theme in the text.

No sub-themes exist for Theme 1, "N/A" is written under the column to represent no sub-themes.

2.22. Theme 6 is listed as "N/A" in the table, but some statements related to future digital tool use are mentioned in the text. To maintain consistency, either include the relevant statements under Theme 6 in the table or adjust the table accordingly if this theme is not relevant to the qualitative

interviews.

The "N/A" came under the column for sub-themes as no sub-themes exist for Theme 6.

Discussion:

2.23. It would be helpful to provide specific references for the studies mentioned in the discussion (e.g., Barakat et al., 2021; Everhart et al., 2021; Fisher et al., 2021; Masoli et al., 2021; etc.). This will allow readers to easily locate and access those studies if they are interested in exploring the topic further.

Thank you for providing these references. This has now been actioned in the discussion.

2.24. Please clarify the link between findings and previous research e.g., when discussing the findings in relation to previous research, it would be beneficial to explicitly state how the current study's findings align or differ from previous research. This will help to establish the contribution and novelty of the current study in the field.

Further clarification has been added to the discussion.

2.25. Expand on the limitations: While the discussion briefly mentions the limitation of not exploring the views of participants/carers, it would be valuable to discuss the potential impact of this limitation on the overall findings and the implications for future research. Additionally, any other limitations of the study should be acknowledged and their potential impact on the interpretation of the results should be discussed

Thanks, further information has now been added regarding the limitation of not exploring the views of participants/carers (page 25-26). "A further limitation of this study is not exploring the views of participants/carers due to limited participant responses. Given that our findings, alongside previous research, suggest that participants/carers are a key user of digital tools, future research should seek to understand their experiences, ensuring there are no barriers to accessing/using digital tools to prevent digital exclusion and promote diversity in participants".

Reviewer: 2

Dr. Mark Jeffries, University of Manchester

Comments to the Author:

BMJ Open Review

Using Digital Tools in Clinical Health and Social Care Research: A Qualitative Study of UK Stakeholders.

Thank you for the opportunity to review this paper which I found interesting to read. This covers an important area of concern for researchers and should be of interest to many involved in health and clinical research.

I have a few points for consideration by the authors in revising the manuscript. I hope they are of some help.

Title and general point

3.1. I think the title is inaccurate considering that half the paper is survey data, some of which quantified and presented as numbers and percentages. I would suggest that this should be a mixed methods study.

The title has now been updated to reflect the mixed methods used, see editor comment 1.4.

3.2. I find the scope of the paper very broad, covering many different types of digital technology which possibly makes specific and tailored recommendations difficult. There is also a huge difference between the use of large datasets for cross sectional epidemiological studies, digital technologies for clinical trials and digital technologies used in qualitative data collection. I'm unsure whether the differences and distinctions between these are brought out in the paper. Essentially there's not much that can be done about the breadth, since that's what you set out to find in some ways, and what your study was about, so I'm not suggesting a narrowing of focus, but I do think that scope and breadth needs greater acknowledgment and discussion both in the introduction and discussion sections.

Further information has now been added to both introduction (page 3) and discussion (page 25).

Abstract

3.3. Could you please explain the reasoning behind the decisions to boundary recruitment to one geographical area and to the last two years?

One geographical area was focused on due to due to availability of data and contact details for clinical research supported by the funder (NIHR CRN West Midlands).

This information has now been added to the abstract (page 1); “To explore the digital tools used since the pandemic, in both online surveys [Researchers and Related Staff (n=41), Research and Development staff (n=25)], participants needed to have worked on clinical, health or social care research studies over the past two years (2020-2022) in an employing organisation based in the West Midlands region of England. This study was funded by the Clinical Research Network NIHR West Midlands and therefore focused on participants based in this region”.

Strengths and Limitations bullet points

3.4. You highlight as a strength that the survey and qualitative questions were co-designed with an experienced stakeholder group. I can't find, and I may have just missed, this mentioned in detail in the body of the manuscript. Could you please provide details of who this stakeholder group was and details of how the questions for the survey and interviews were developed.

This information has now been added into the manuscript on page 4, “The survey and qualitative questions were co-designed with an experienced stakeholder group consisting of three individuals working in senior positions within clinical research based in the West Midlands. The stakeholder group met once during the project on Microsoft Teams (February 2022), the research team presented an overview of the project and survey questions and the steering group offered verbal feedback. The stakeholder group members were emailed a draft of qualitative interview materials in August 2022 and provided feedback via email on the qualitative questions”

Introduction

3.5. I feel the introduction needs more detail and more background including referencing a greater range of previous literature. There's somewhat a lack of clarity here. You mention, in parentheses, a number of different digital tools and do the same when discussing how these are used in different stages of research. In the second paragraph on page 5 you don't mention data collection tools but then do in the aims of the study just below. Could you provide, maybe in a table, as a suggestion, the sorts of digital tools used and at the various stages in research they are used. I feel this section requires greater clarity to enable the reader to fully understand why this is important and how specifically, the roles of different digital tools in research have been previously understood.

Data collection has now been added on page 3. The introduction was left purposefully general as one of the aims of the study was to investigate which tools have been used, we didn't want to pre-empt the results of study. Given the limited word count and the fact our study is a mixed-methods study and includes qualitative results/quotes, we were restricted by these two factors and therefore we wanted to achieve a concise introduction.

3.6. Related to the above regarding the different research stages in which digital tools can be used you mention only one previous piece of research here relating to recruitment and retention but nothing relating to other stages of research. Other research about other stages might help – for instance Shamsuddin et al on Online Workshops during Covid 19 might be useful (I am not an author of that paper so have no stake!) I think adding further literature here would be useful in contextualising how your study aimed to build upon that.

The reference to online workshops has now been incorporated into the introduction. The piece of research specifically on recruitment and retention has been highlighted as this research study investigated stakeholder's experiences when using digital tools. Given this is also the focal point of our study, this particular study was highlighted as we wish to build on these findings but approach the

topic area with a broader scope of research stages. Amendments have been made to page 3-4 to make this more evident.

Methods

3.7. Generally I feel the methods are really detailed and clear.

Thank you.

3.8. Consent for the surveys – was it embedded in the Qualtrics platform or sent as an additional document?

For the online surveys, consent was embedded in the Qualtrics platform (page 7), see response to comment 2.12.

3.9. Page 7 line 14 – Typo missing apostrophe “participants’ experience”

This has been amended (page 6).

3.10. Page 7 Lines 15-16 section in brackets – not clear what you mean - please expand.

This information has now been added into the text on page 8; “Version 12, QSR International’ is referring to the version of NVivo we used for analysis along with details of the owning company”.

3.11. Data saturation. I agree with your definition here but exactly how did you decide no further themes could be interpreted?

Reference to data saturation has now been removed. See response to reviewer comment 4.5.

3.12. If you are going to put Thematic Analysis in capitals you might need to cite Braun and Clarke

Reference has now been added.

3.13. Great detail in the analysis section but maybe this also needs something on how you moved from the final coding to the interpretation of themes. Feels that there might be a step missing here.

Further detail has been now added, see response to reviewer comment 2.14.

Results

3.14. Overall, I think these are really interesting, with valid and interesting themes and they read really well.

Thank you.

3.15. 19/52 completed all the first survey and 22/25 the second. Can you please, somewhere, reflect upon these numbers and whether you felt they were sufficient for the interpretations you make?

The survey was intended as a way to recruit participants into the qualitative interviews and to gain a general understanding about tools people are using. Reference to the small participant numbers have now been added to the discussion as a limitation.

3.16. Similarly, as I understand, and I may have this wrong, each digital tool is mostly only mentioned once. Does that tell us much about their use?

This has now been added to the discussion, page 25; “However, a limitation of this approach is the wide scope of the survey questions (i.e. covering all research stages) which does not allow deeper exploration into specific tools. It was apparent that many of the tools identified within the surveys were only referenced once by a single participant, suggesting a possible lack of consistency in the tools used within clinical research and the need for standardisation across each research stage. Alternatively, another explanation could be the small number of survey respondents which could make it difficult to generalise the results. Future research could elaborate on this work and investigate tools used in different geographical regions”.

3.17. Table 5 is really useful but I feel that most qualitative studies also embed some quotations in the text. It is somewhat difficult to jump back and forward from the text in the results to the table.

A strict word-count set by the journal has required the quotes to be collated in a table.

Discussion

3.18. Para 2 Page 23. This relies quite heavily on one previous study. Is there anything else from previous literature that your findings build upon?

This study in particular was pivotal to compare and contrast our results to due to its similar nature. The study in question investigated participant’s experiences of using digital tools in clinical research, however our study has a broader scope. We have now incorporated more references into the discussion.

3.19. Your findings and considerations around inclusivity are really interesting and important.

Thank you.

References

3.20. Related to some of my points above. You draw upon and cite only 14 references. I would have expected a study such as this with this sort of scope to draw upon a wider literature.

Further references have now been added to the introduction and discussion sections.

PPI

3.21. You had no PPI involvement. Could you reflect on why this was the case and perhaps discuss this as a limitation?

A third survey formed part of this study, aimed at collating the experiences of participants and carers of digital tools. However, this was not included in the data analysis due to lack of participant responses. This has now been included in the methods section as well as a limitation in the discussion.

Reviewer: 3

Dr. David Wyatt, King's College London

Comments to the Author:

Thank you for giving me the opportunity to review this interesting paper, which draws on surveys and semi-structured interviews to explore which digital tools are used in clinical research, the experiences of a broad range of staff of these tools and their effectiveness.

The authors present some rich data (particularly in Table 5), and I think this paper has a lot of potential and is of value to readers of BMJ Open. In its current form, it has some flaws and I hope the comments I provide below help the authors to refine this paper for publication.

Introduction

4.1. At the moment, the reader is not provided with any definition or understanding of how the authors are operationalising 'digital tools' in this paper. I think this needs to be added (even if just using the definition provided to survey participants in Table S1).

A definition has now been added into the introduction, page 3, "Digital tools can be defined as an alternative to paper-based methods that is IT based or an online platform that aids any aspect of the research study".

4.2. The introduction also mentions the effectiveness of tools as one of the aims of the study (absent from the Strengths and limitations bullets or the paper's Discussion. I can see that this was covered in the survey data which is currently in a supplementary table. If effectiveness is going to remain, it needs to be discussed in some detail in the paper itself (although it might be easier just to remove effectiveness as there's already a lot here to cover).

Reference to effectiveness has now been removed from the aims of the study.

Methods

4.3. While the methods section is detailed, there is not a single reference to methods literature. References has now been added (e.g., Braun and Clarke) as well as academic literature supporting the use of Qualtrics in research studies.

4.4. Surveys - In the Strengths and weaknesses bullet points, the authors mention that the surveys were co-designed with an experienced stakeholder group. A little detail about this process would be really valuable here. This is particularly important because later on page 8 the authors mention a selection from a dropdown list of reasons for not using digital (which I assume was part of the co-design process).

Further information relating to the stakeholder group can be found in response to comment 3.4.

4.5. 8 interviews is a very small number to claim data saturation, particularly when the number of digital tools available and their potential uses are so broad, and you had two very large staffing groups (R&D staff (n 5)- there are so many roles here that link to research, and Researcher and related staff (n3) - from CRNs, CRPs, university research staff, trial managers etc. My gut feeling is that it might be better to talk about these interviews as a means of adding further texture to your survey data (which is what I understood was the purpose of them anyway) and leave it there.

If the authors want to make this claim of data saturation, it would be good to add a sentence or two explaining how exactly they decided that saturation was met (so that this claim is a little more robust). For example, I assume you were analysing as you went along, so you might have done 6 interviews, realised you were getting similar things, and then been more aware of new points (or the lack thereof) in the last 2 interviews. The team might have looked at the data and agreed no new codes were identifiable.

Thank you for this suggestion, on reflection, we agree with this comment and reference to data saturation has now been removed.

Results

4.6. The survey results at the moment have a short paragraph and a table per survey. There is a lot of data here to process, and some rather small numbers. It would be good to add one or two more paragraphs signposting the reader to particular pieces of data you want them to focus on.

Some further detail has been added. However due to the strict word limit of the journal, we have focussed on discussing the more detailed qualitative results.

4.7. Are the authors able to give any sense of how many people each of the surveys were sent to/the response rates? The numbers are quite small so any context here would be useful to know.

This has been addressed in previous comments above, see response to reviewer comments 2.6.

Thematic analysis of qualitative data

4.8. I struggle a little with this thematic analysis because the themes match so neatly on the question topics listed in Table S3. As such they're really quite descriptive rather than analytic. While these broad topics/headings seem useful in structuring the data, I think a deeper look at some of the points within it would be useful. For instance, on page 17 theme 2 "impacts of the Covid-19 pandemic", the last sentence is the most interesting point here (in my opinion) - not that covid-19 accelerated the use of digital (this has been said before). But that Covid provided a way to, as Table 5, point 2.2 says, "proved that the digital way is working" so that it remains in routine practice and continues to be a space of innovation. Could this point be developed more? (if appropriate)).

Further detail has been added to theme 2, page 19. The title has been changed; "COVID-19 Pandemic: Validating the benefits of working digitally". Further detail has been added to the text "The pandemic served as a trial period, to prove that "the digital way is working", supporting continued digital tool use and innovation in routine practise. There was a sense of resistance from participants to not revert back to seemingly less efficient non digital methods (e.g., paper questionnaire compared to online)".

4.9. Table 5, Theme 3, Drawbacks - an interesting theme that appears for me here is one of connection, both in the human and the WIFI sense. Another theme is of inequalities. I obviously am only seeing a snapshot of the data in Table 5, but it seems that revisiting the data and thinking beyond the existing themes and more analytically would be useful.

Thank you for this suggestion, further detail has been added to theme 3, in the section called 'drawbacks' on page 20; "Some of the drawbacks of digital tools referenced were of a practical nature,

for example technical issues such as poor internet connection or researchers may require additional technical support. However, it was noted that face-to-face research studies may also require additional support. Other drawbacks related to connection, specifically difficulty reading interactions online” and “Participants raised concerns around inequalities and lack of inclusivity relating to digital tool use”.

Discussion

4.10. The discussion summarises the paper well and presents some implications for the future. I think some of this text will need to be updated if the authors revisit the analysis.

The discussion has been revised and updated as required.

VERSION 2 – REVIEW

REVIEWER	Wyatt, David King's College London, Division of Health & Social Care
REVIEW RETURNED	03-Oct-2023
GENERAL COMMENTS	Thank you for carefully amending your interesting paper. I look forward to seeing it in print!